# B Cell Activating Factor Induces Drug Resistance in Hairy Cell Leukemia Variant

**DOI:** 10.3390/biomedicines13040890

**Published:** 2025-04-07

**Authors:** Claire Fritz, Daniel Feinberg, Akshaya Radhakrishnan, Kayla Klatt, E. Ricky Chan, Philip Rock, Richard Burack, Reshmi Parameswaran

**Affiliations:** 1Department of Pathology, Case Western Reserve University, Cleveland, OH 44106, USA; cef78@case.edu (C.F.); daf93@case.edu (D.F.); krk110@case.edu (K.K.); 2Division of Hematology/Oncology, Department of Medicine, Case Western Reserve University, Cleveland, OH 44106, USA; axr1066@case.edu; 3Department of Population and Quantitative Health Sciences, Case Western Reserve University, Cleveland, OH 44106, USA; erc6@case.edu; 4Department of Pathology and Laboratory Medicine, University of Rochester Medical Center, Rochester, NY 14642, USA; philip_rock@urmc.rochester.edu (P.R.); richard_burack@urmc.rochester.edu (R.B.); 5Pediatric Hematology and Oncology, The Angie Fowler Adolescent & Young Adult Cancer Institute, University Hospitals Rainbow Babies & Children’s Hospital, Cleveland, OH 44106, USA; 6The Case Comprehensive Cancer Center, Case Western Reserve University School of Medicine, Cleveland, OH 44106, USA

**Keywords:** hairy cell leukemia, chemoresistance, BAFF, HCL, Leukemia, chemo-protection, drug resistance

## Abstract

**Background:** Chemoresistance is an existing challenge faced in the treatment of the hairy cell leukemia variant (HCL-v). Classical hairy cell leukemia (HCL-c) is very sensitive to the standard of care with purine nucleoside analogs (PNAs) cladribine (cDa) and pentostatin. However, almost half of these patients eventually become less sensitive to chemotherapy and relapse. HCL-variant (HCL-v) is a biologically distinct entity from HCL-c that is not sensitive to frontline PNA therapy, and this treatment is not recommended for these patients. To address these treatment challenges, we investigated the role of B-cell activating factor (BAFF) in promoting HCL-v cell chemoresistance. **Methods:** Flow cytometry and quantitative PCR were used to measure the levels of BAFF and its receptors. To determine BAFF activated pathways in HCL-c and HCL-v, the Bonna-12 HCL-c cell line or HCL-v patient-derived cancer cells were stimulated with recombinat BAFF and activation of common BAFF-activated pathways, including the nonclassical nuclear factor kappa B (NF-κB) pathway, the Extracellular Signal-Regulated Kinase (Erk) and phosphatidylinositol-3 (PI-3) kinase (PI3K)/AKT serine/threonine kinase (AKT) pathways were measured by western blotting. To test whether BAFF signaling promotes chemoresistance in HCL-v, we stimulated patient-derived HCL-v cells with BAFF and performed RNA sequencing. Lastly, to confirm the functional implications of BAFF signaling in HCL-v, we treated patient-derived HCL-v cells with exogenous BAFF before treatment with cladribine. **Results:** We found that HCL-v patient-derived cancer cells express receptors of BAFF at varying degrees and express relatively lower levels of membrane-bound BAFF ligand expression. BAFF stimulation of these cells resulted in substantial activation of the nonclassical NF-κB pathway, which is known to promote anti-apoptotic and pro-survival effects in B-cell cancers. Conversely, in the Bonna-12 cell line, we observed constitutive activation of the nonclassical NF-κB pathway. Through RNA sequencing, we found that BAFF upregulates a myriad of genes that are known to promote chemoresistance in various cancers, including *IL1*, *CXCL1/2*, *CXCL5*, *CXCL8*, *TRAF3*, and *PTGS2.* Lastly, we found that BAFF protects these cells from cladribine-induced cell death in vitro. **Conclusions:** We conclude that BAFF provides chemo-protection in HCL-v cells by activating nonclassical NF-κB signaling, which results in the upregulation of multiple pro-survival or anti-apoptotic genes. Our results highlight an important role of BAFF in HCL-v resistance to chemotherapy and suggest that the BAFF blockade may enhance the chemosensitivity to PNAs in drug-resistant HCL-v patients.

## 1. Introduction

Classical hairy cell leukemia (HCL-c) is a rare malignancy of mature B-cells that makes up 2% of mature non-Hodgkin lymphomas (NHL) [1]. HCL-c is characterized by the presence of mature small B lymphoid cells, with distinctive circumferential cytoplasmic villi, or “hairy” projections, in the peripheral blood, bone marrow, and splenic red pulp [1,2]. This underlies the clinical symptoms seen in HCL patients, which include splenomegaly, pancytopenia, anemia, neutropenia, thrombocytopenia, and infection [1]. Diagnosis involves the detection of the “hairy” leukemic cells in the peripheral blood and the expression of tartrate-resistant acid phosphatase (TRAP), CD103, CD123, CD25, and CD11c [3,4,5]. The BRAF^V600E^ point mutation, leading to constitutive mitogen-activated protein kinase (MAPK) signaling, is seen in up to 90% of cases and is thought to be a driving genetic event in HCL pathogenesis [6].

HCL-c is an indolent cancer that follows a chronic disease course. The standard of care for symptomatic HCL-c patients includes a combination of rituximab with the purine nucleoside analogs (PNAs) cladribine (cDa) or pentostatin [4,7]. Although treatment with PNAs is effective, up to 58% of patients relapse within five years of initial treatment and become less responsive to therapy [8,9]. Further, retreatment results in increased toxicity and cost without evidence of a cure [10,11,12]. For these reasons, the current HCL-c research is focused on identifying novel therapeutic targets to limit minimal residual disease (MRD) and prolong relapse-free survival [9,13]. The treatment strategies under investigation include BRAF/MEK inhibition, Bruton’s tyrosine kinase (BTK) inhibition, and CD22-directed therapies [7,8]. Ongoing investigations into HCL-c disease biology will allow for the development of novel therapeutic combinations to enhance chemosensitivity and prevent relapse.

The variant form of hairy cell leukemia (HCL-v) has overlapping clinicopathological features with HCL-c but is biologically distinct and presents unique treatment challenges [14]. The 5th edition of the World Health Organization Classification of Hematolymphoid Tumor classifies HCL-v as splenic B-cell leukemia with prominent nucleoli (SBLPN) [15]. The circulating HCL-v cells have a morphology in between prolymphocytes and hairy cells [16]. These cells lack the expression of TRAP and the BRAF^V600E^ mutation and have a low immunogenic score, with no CD25 expression and weak CD123 expression [3,17]. Further, HCL-v patients have unique clinical presentations. These patients do not present with splenomegaly, leukocytosis, or infections. The HCL-v disease course is significantly more aggressive than HCL-c, with a median overall survival of 9 years and a lack of response to PNAs [16]. The response rates to the combination of cDa and rituximab and to B-cell receptor (BCR) signaling pathway inhibitors are also inferior to HCL-c patients [16]. In addition to the lack of response to chemoimmunotherapy and lack of molecular therapeutic targets, there is limited data from multi-center studies due to the rarity of the disease, making treatment guidelines more difficult.

Enhancing the chemosensitivity of classical and variant HCL cells would help to improve the outcomes of both diseases. This could help prevent relapse following PNA use for HCL-c patients and make frontline treatment more effective for HCL-v patients. To this end, we are interested in identifying signaling pathways and genes that may be promoting chemoresistance in HCL-v cells. B-cell activating factor (BAFF) is a tumor necrosis factor (TNF)-family ligand that plays a critical role in promoting the maturation and survival of developing B-cells [18]. Throughout their developmental processes, B-cells express the three receptors of BAFF: BAFF receptor (BAFF-R), B-cell maturation antigen (BCMA), and transmembrane activator and calcium-modulator and cyclophilin ligand (CAML) interactor (TACI) [19,20]. BAFF ligation to these receptors activates pro-survival and pro-growth pathways, including the classical and nonclassical nuclear factor kappa B (NF-κB) pathways and the extracellular signal-regulated kinase (ERK) and phosphatidylinositol-3 (PI-3) kinase (PI3K)/AKT serine/threonine kinase (AKT) pathways, to promote development and prevent premature cell death [18,19,21].

Multiple cancers are known to rely on BAFF signaling to gain a survival edge. High levels of serum BAFF and its receptors are seen in various B-cell cancers and are associated with disease aggressiveness and a lack of therapeutic response [22,23,24,25]. Similarly, multiple in vitro studies have demonstrated that BAFF stimulation reduces the chemosensitivity of cancer cells, including multiple myeloma, chronic lymphocytic leukemia, and mantle cell lymphoma cells [26,27,28,29]. Conversely, BAFF neutralization or silencing restored chemosensitivity to various agents. Therefore, we hypothesized that BAFF may have a role in promoting chemoresistance in HCL-c and HCL-v and that its neutralization may overcome treatment barriers, including the lack of initial response and relapse.

To this end, we embarked on exploring the role of BAFF in HCL-v chemoresistance. We demonstrate that HCL-c cell lines and HCL-v patient samples express BAFF receptors, most significantly BAFF-R and TACI. Because BAFF ligation to BAFF-R most significantly activates the pro-cancerous nonclassical NF-κB pathway, we explored whether BAFF may activate this pathway in HCL-c and HCL-v cells. We found constitutive activation in the Bonna-12 HCL-c cell line and BAFF-induced signaling in HCL-v patient-derived cells. We saw selective activation of this pathway, as the ERK and AKTpathways were not significantly activated by BAFF in HCL-v cells. Through RNA sequencing, we found that BAFF induces the expression of a host of genes known to promote chemoresistance. Therefore, we explored whether BAFF can directly induce chemoprotection in vitro in HCL-v cells. Functionally, we found that BAFF stimulation protected HCL-v patient-derived cells from chemotherapy-induced cell death.

## 2. Materials and Methods

### 2.1. Patient Samples

All HCL-v patient samples were provided by the Hematological Malignancy Procurement Program of the Wilmot Cancer Institute at the University of Rochester Medical Center. These are splenic lymphocytes obtained by the Surgical Pathology Department after diagnostic biopsies. These discarded and de-identified patient samples were obtained under an approved IRB exemption protocol. All patients had an HCL-v diagnosis and expressed the markers CD11c, CD19, CD20, and CD22.

### 2.2. Cell Lines

The Bonna-12 (Item#: ACC 150) and Hair-M (Item#: ACC 762) cell lines were obtained from the Leibniz Institute DSMZ—German Collection of Microorganisms and Cell Cultures. Both cell lines were originally established from classic hairy cell leukemia patients.

### 2.3. Flow Cytometry

For flow cytometry staining, cells were washed with PBS, resuspended in 50 µL PBS, and stained with 1.5 µL of the following antibodies: PE BAFF-R (Biolegend, San Diego, CA, USA, catalog # 316906, clone 11c1), PCY-7 TACI (Biolegend, catalog # 311908, clone 1A1), Percp BCMA (Biolegend, catalog # 357509, clone 19F2), and APC BAFF (Biolegend, catalog # 366508, clone 1D6). After 15 min of incubation at room temperature in the dark, 1 mL of PBS was added to wash out unbound antibodies before centrifugation. Cell pellets were resuspended in 120 µL of PBS and run on the CytoFLEX flow cytometer (Beckman Coulter, Brea, CA, USA). Live cells were gated from dead/dying cells by size on forward/side scatter dot plots for each sample. Analysis was performed on CytExpert software version 2.5.0.77.

### 2.4. Real-Time Quantitative PCR

RNA was isolated from cell lines or patient-derived samples using the Qiagen RNAEasy kit (catalog # 74104). The RNA concentration and purity (260/280 nanometer ratio 2.0–2.1) was measured via NanoDrop™ 2000/2000c spectrophotometers (Thermo Scientific™, Waltham, MA, USA), and 1 µg of RNA was reverse-transcribed into cDNA using the iScript^TM^ Reverse Transcription Supermix (Bio-Rad, Hercules, CA, USA, catalog# 1708840). The reaction mix was loaded into the Bio-Rad T100 thermocycler with the following protocol: 5 min at 25 °C, 20 min at 46 °C, and 1 min at 95 °C, according to the manufacturer’s instructions. For qPCR, 1 µg of cDNA was loaded with nuclease-free water, primers (see Table 1 for sequences), and iQ Sybr green supermix (Bio-Rad, 170880). The manufacturer’s protocol was followed for the cycling conditions and analysis.

### 2.5. Western Blotting

Cell lines or patient samples were stimulated with 250 ng/mL of recombinant human BAFF (rhBAFF) (Biolegend, 559606) for 12–16 h. Total protein lysate was extracted with RIPA buffer. Cells were washed with PBS, and the cell pellet was resuspended in 70 µL of RIPA buffer with fresh protease inhibitors (Sigma-Aldrich, St. Louis, MO, USA, PPC1010). Samples were placed on a rotator for 40 min and were then centrifuged at 16,000 RCF for 15 min at 4 °C. The protein lysate was saved. For the cytoplasmic nuclear fractionation, the NE-PER™ nuclear and cytoplasmic extraction reagents were used (Thermo Fischer Scientific, catalog# 78833) following the manufacturer’s protocol. The protein concentration was quantified by the Bradford Assay (Pierce BCA Protein Assay Kit with Dilution-Free BSA Protein Standards, catalog# A55864, Thermo Fisher Scientific) and normalized across the samples. Samples were combined with 4× Laemmli sample buffer (Bio-Rad, catalog# 1610747) and β-mercaptoethanol, boiled, and loaded into a 10% agarose gel. After transfer to 0.4 µm PVDF membrane, it was blocked with 5% milk in TBS-T for 1 h. Primary and secondary antibody incubations were 1 h at room temperature or overnight at 4 °C. Membranes were washed with 1× TBS-T three times for five minutes after antibody incubations. The following primary antibodies were used (all from Cell Signaling Technology (Danvers, MA, USA)): p100/p52 NF-κB (catalog# 4882T), lamin B2 (catalog# 12255S), α-tubulin (catalog# 3873S), GAPDH (catalog# 97166S), phospho-ERK (catalog# 9101S), ERK(catalog# 4695S), phospho-AKT (catalog# 4058T), and AKT (catalog# 4691S). The β-actin (sc-47778) primary antibody was obtained from Santa Cruz Biotechnology (Dallas, TX, USA). Secondary antibodies were purchased from Cell Signaling Technology: anti-rabbit (7074S) and anti-mouse (7076S). Membranes were developed with Clarity^TM^ Western ECL substrate (Bio-Rad, catalog# 1705061) or Amersham™ ECL Select™ Western Blotting Detection Reagent (Cytiva, Marlborough, MA, USA, catalog# RPN2235) and were imaged on the Bio-Rad ChemiDoc. Quantification of the band intensity was performed using ImageJ software version 1.54d. For normalization, the intensity of the experimental band was divided by that of the loading control GAPDH. For the phosphorylation studies the intensity of the phosphorylated form of ERK or AKTwas divided by the total protein levels and then divided by the GAPDH intensity level for normalization. To investigate the reliance on the BAFF receptor for BAFF-induced activation of the nonclassical NF-κB, the cells from Patient 131 were pre-incubated with 500 ng/mL of recombinant human BAFF alone or in the presence of 15 µg/mL of BAFF receptor-neutralizing antibody for 16 h. Cytoplasmic-nuclear fractionation and probing of the nonclassical NF-κB pathway was performed as described above.

### 2.6. RNA Sequencing

HCL patient-derived splenocytes were stimulated with 250ng/mL recombinant human BAFF (Biolegend, catalog 559606) for 16 h. RNA was isolated using the Qiagen RNAEasy extraction kit (catalog, 74104). Using the NanoDrop™ 2000/2000c Spectrophotometers (Thermo Scientific™), the RNA concentration and purity were measured. RNA was normalized to 50 ng/µL in nuclease-free water and submitted to the CWRU Genomics Core for quality control measurements and sequencing. RNA-Seq libraries were sequenced using the Illumina platform (Novogene Corporation Inc., Sacramento, CA, USA). Reads were trimmed and assessed for quality using TrimGalore! v0.4.2 (Babraham Bioinformatics, Cambridge, UK), which is a wrapper script incorporating FastQC and cutadapt. Filtered reads were aligned to the human reference genome (GRCh38) with STARaligner v2.7.0. Annotation of the aligned reads was done using the GENCODE annotation for GRCh38. Differential expression at the gene level was assessed using Cufflinks v2.2.2, and expression values are reported as fragments per kilo base of exon per million fragments mapped (FPKM). Differentially expressed genes were identified using a significance cutoff q-value < 0.05 using Benjamini–Hochberg multiple testing correction. The results of the differential expression were further analyzed using iPathwayGuide (AdvaitaBio, Ann Arbor, MI, USA) for pathway analysis.

### 2.7. Chemoresistance Experiment

Splenocytes derived from an HCL patient (Patient 058) were used for chemoresistance experiments, and 500,000 cells per well in 1 mL of complete RPMI were plated in a 24-well plate. Cells were pre-incubated with 250 ng/mL (1 µL) of rhBAFF (Biolegend, catalog# 559606) for 3 days. After 3 days, the cells were treated with 250 nM (1 µL) of cladribine (APExBIO, Houston, TX, USA, 501904737) and repeated BAFF stimulation. Control wells received an equal volume of the solvent 0.1% DMSO. Cells were stimulated with BAFF again on day 5. Cell number and viability were measured daily by trypan blue staining (1:1 volume of stain:cells) and Countess 3 Cell counter (Thermo Fischer).

### 2.8. Statistics

Statistical tests are described in the respective figure legends. All statistical tests were performed using GraphPad Prism 10.2.0. For RNA sequencing analysis, differential genes were identified with a significance cutoff q-value < 0.05 using Benjamini–Hochberg multiple testing correction. For the functional chemoresistance studies a one-way ANOVA was used to compare the ‘BAFF’ and ‘BAFF + cDa’ groups over time. Tukey’s multiple hypothesis correction was used.

## 3. Results

### 3.1. HCL-c and HCL-v Cells Express Receptors of BAFF

We first checked whether HCL-v patient samples and HCL-c cell lines express the receptors of BAFF and membrane-bound ligand BAFF. In the cell lines and patient samples that we analyzed, we saw a heterogeneous pattern of expression of membrane-bound BAFF and its receptors. Significant populations of cancer cells from HCL-v Patients 030, 058, 131, 154, and 195 expressed BAFF-R (68.30%, 51.1%, 77.8%, 10.0%, and 63.0%) and TACI (43.8%, 39.7%, 75.2%, 13.6%, and 5.6%), respectively, per flow dot plots (Figure 1A). However, we did not see a distinct BCMA-positive population in these samples. Cancer cells from patients 058 and 154 had discernable membrane-bound BAFF-expressing cells, although this was not as highly expressed as the receptors. The expression patterns of BAFF and its receptors were validated at the gene level by RT-qPCR. Jeko-1 cells, a mantle cell lymphoma line, were used as our reference, because these cells are known to express BAFF and its receptors and depend on BAFF survival signaling [26]. The HCL-c cell lines Bonna-12 and Hair-M had a greater gene expression level of BAFF, TACI, and BCMA relative to the Jeko-1 control and relatively lower levels of BAFF-R expression.

### 3.2. BAFF Preferentially Induces Signaling Through the Nonclassical NF-κB Pathway in HCL-v Cells

After confirming that HCL-c/v cells express BAFF, we were interested in determining pathways that may be activated in HCL-c and HCL-v cells upon BAFF stimulation. The main pathways that BAFF is known to activate include the classical and nonclassical NF-κB pathways and the ERK and PI3K/AKT pathways [30,31]. The nonclassical NF-κB pathway is most efficiently activated with BAFF ligation to BAFF-R relative to TACI or BCMA. Upon ligand binding and receptor activation, the central MAP kinase NF-kappa-B-inducing kinase (NIK) is stabilized, leading to activation of IKKα and proteolytic cleavage of p100 to p52. P52 then dimerizes with RelB and translocates to the nucleus to activate transcription [32]. Pathway activation, as measured by the accumulation of nuclear p52, was independent of exogenous BAFF stimulation in Bonna-12 cells, suggesting an autocrine BAFF signaling and constitutive pathway activation in these HCL-c cells (Figure 2A). Conversely, we observed a clear exogenous BAFF-induced accumulation of nuclear p52 in HCL-v cells (Patient 131), suggesting that these cells depend on paracrine BAFF signaling (Figure 2B). BAFF-induced nonclassical NF-κB pathway activation was also seen in the total protein lysate from another HCL-v patient sample, Patient 154 (Figure 2C, left), suggesting that this finding might be conserved among HCL-v patients. We also tested other BAFF-induced pathways in these patient cells: the ERK and AKT pathways. Activation was measured by increased phosphorylated-ERK or AKT, respectively. In Patient 154, we saw a modest increase in phosphorylated ERK but decreased phosphorylated AKT (Figure 2C, right). Therefore, we conclude that the pro-survival effects of BAFF stimulation of HCL cells are likely mediated most significantly by the nonclassical NF-κB pathway. We then stimulated patient cells with BAFF in the presence of a BAFF receptor-neutralizing antibody and found a drastically decreased activation of the nonclassical NF-κB pathway relative to the BAFF-stimulated control. Thus, we conclude that specific activation of the BAFF receptor induces activation of this pathway in HCL-v cells.

### 3.3. BAFF Induces Multiple Genes Involved in Chemoresistance in HCL-v Cells

The expression patterns among the control and BAFF-treated groups in the principal component analysis (PCA) showed distinct clustering between the control and BAFF-stimulated samples (Figure 3A). There were 1447 differentially expressed (DE) genes with BAFF stimulation among 15,008 with measured expression. The volcano plot displays only the significant (*p*-value < 0.05, fold change > 0) differentially expressed genes (Figure 3B). The top biological processes that were activated with BAFF stimulation include cytokine–cytokine interaction (53 genes, 1.120 × 10^−8^), MAPK signaling (42 genes, 2.659 × 10^−8^), NF-κB signaling (27 genes, 6.527 × 10^−7^), and pathways in cancer (71 genes, 1.411 × 10^−6^) (Figure 3C). The most significantly differentially expressed genes upon BAFF stimulation with each biological process are mapped in Figure 3C.

Notably, many of the genes upregulated among these processes have been implicated in chemoresistance in various cancer models. Interleukins alpha and beta (*IL1A* and *IL1B*) were among the most differentially upregulated gene products with BAFF stimulation (Log_2_FC 6.4 and 4.6, *p*-values 0.001 and 0.001, respectively). IL-1 signaling has been increasingly implicated in chemoresistance in multiple solid tumors, including breast, gastric, non-small cell lung cancer, head and neck squamous cell carcinoma, and colorectal cancer [33,34,35,36,37,38]. For example, IL-1β was found to mediate the resistance of non-small cell lung cancer cells to docetaxel by regulating the formation of polyploid giant cancer cells, while its inhibition increased docetaxel sensitivity. As such, anakinra, an IL-1 receptor (IL1R1) antagonist, is being or has been investigated in combination therapies for many of these cancers to enhance chemosensitivity [39]. Although the data are less mature, IL-1 chemoresistance is being investigated in hematological malignancies as well, including chronic myeloid leukemia (CML) and multiple myeloma [40]. In preclinical CML models, adding a recombinant IL1 receptor antagonist increased the sensitivity of leukemic stem cells to BCR-ABL tyrosine kinase inhibitor [41].

In addition to IL-1 upregulation, BAFF stimulation increased the expression of many chemokines and chemokine receptors, which were classified into the biological processes ‘cytokine–cytokine receptor interactions’ and ‘pathways in cancer’. These include *CXCL1* (Log_2_FC 2.515, *p*-value 0.001), *CXCL2* (Log_2_FC 2.461, *p*-value 0.001), *CXCL8* (Log_2_FC 1.923, *p*-value 0.001), and *CXCR5* (Log_2_FC 2.021, *p*-value 0.001), all of which promote chemoresistance (Figure 3B and Figure 4) [42,43,44]. For example, in breast cancer models, treatment with chemotherapy was found to induce TNF-α activation of NF-κB to increase *CXCR1/2* expression, to reduce sensitivity to therapy [42]. *CXCL8* was also identified in triple-negative breast cancer as a potential druggable target, as it was found to promote paclitaxel resistance [44].

In addition, 71 differentially expressed genes were grouped into ‘pathways in cancer’, and a number of these genes are known to promote chemoresistance. Prostaglandin-endoperoxide synthase-2 (*PTGS2*, Log_2_FC 5.416, *p* = 0.001) was the most highly expressed gene in this group with BAFF stimulation and is associated with chemoresistance and predicts poor survival in non-small cell lung cancer (NSCLC) (Figure 3D and Figure 4D) [45]. Mechanistically, cisplatin induces *PTGS2* expression through ERK1/2-NF-κB signaling, which increases *PTGS2* in a positive feedback loop and promotes multidrug resistance [45]. *TRAF1* was also upregulated with BAFF stimulation in HCL cells and is also identified as an attractive therapeutic target in multiple cancers (Figure 3D and Figure 4D). TRAF1 confers resistance to apoptosis in NSCLC, Hodgkin’s lymphoma, and renal cell carcinoma, in which it specifically diminishes the sensitivity to sunitinib [46,47,48].

### 3.4. BAFF Promotes Chemoresistance in HCL-v Cells

After we observed that BAFF upregulated many genes that are implicated in chemoresistance in HCL cells, we sought to verify that these transcriptional changes have a functional effect. To this end, we pre-incubated patient-derived HCL-v cells from Patient 058 with recombinant BAFF before treatment with cDa, the frontline standard of care for classical HCL patients. As expected, cDa alone significantly decreased viability and cell numbers after 3 and 4 days, respectively (Figure 5). However, BAFF rescued these cells from chemotherapy-induced cell death. The cells that were stimulated with BAFF before cDa treatment maintained similar cell numbers and viability as the control or BAFF-only cells, suggesting that BAFF signaling plays a key role in HCL-v chemoresistance. Importantly, the cancer cells used in this study were from an HCL-v patient (Patient 058), and generally, these patients do not respond well to frontline therapies, suggesting that BAFF neutralization could potentially sensitize these patients to frontline chemotherapy.

## 4. Discussion

Resistance to frontline therapy is still a challenge in managing HCL-c and HCL-v patients [7,49]. In this study, we identified BAFF as one of the factors involved in the chemo-protection of HCL-c and HCL-v cells. We verified that HCL-v cells derived from multiple patients express BAFF-R and TACI even at higher levels than the mantle cell lymphoma cell line (Jeko-1), which is known to express these receptors and confer survival benefits from BAFF signaling [26]. HCL-v patient cells expressed both ligand BAFF and its receptors, which points to a possibility of both autocrine and paracrine BAFF signaling happening in these cells. The HCL-c cell lines Bonna-12 and Hair-M showed expression of BAFF, TACI, and BCMA.

We were interested in identifying the signaling cascades induced by BAFF in HCL-c/v cells. In these signaling studies, we found that the HCL-c Bonna-12 cells display constitutive nonclassical NF-κB pathway activation. We attribute this finding to acquired genetic changes in this transformed cell line to gain survival advantage in the culture. This might also be due to BAFF autocrine signaling, as these cells express both BAFF ligand and its receptors. However, in HCL-v patient-derived cancer cells, the nonclassical NF-κB pathway was activated only when stimulated with exogenous BAFF. Our data suggest that activation of this pathway is specifically mediated by BAFF ligation of the BAFF receptor. Persistent signaling of the nonclassical NF-κB pathway upregulates pro-growth and anti-apoptotic functions in multiple B cell cancers, including multiple myeloma, Hodgkin’s lymphoma, mantle cell lymphoma, adult T-cell leukemias/lymphomas, and diffuse large B-cell lymphoma (DLBCL) [32,50,51]. In these cancer models, nonclassical NF-κB pathway activation leads to the upregulation of anti-apoptotic and pro-survival proteins conferring chemoresistance. However, to our knowledge, the BAFF-induced nonclassical NF-κB pathway in HCL-v cells or the potential for this being a therapeutic target have not been described until now.

The PI3K/AKT and ERK pathways are also known to be BAFF-activated in lymphoid malignancies. However, we only saw a modest increase in phosphorylated ERK and a decrease in phosphorylated AKT post-BAFF stimulation in HCL-v patient-derived cells. Additionally, it has been shown that BAFF-R activates PI3K/AKT signaling in human naive B-cells to upregulate the genes involved in migration, proliferation, growth, and survival. However, this does not occur in switched memory cells [21]. It is, therefore, possible that PI3K/AKT signaling downstream of BAFF-R is disease-specific, cell-type specific, or even depends on the maturation stage of the B cells, considering that HCL is a mature B-cell neoplasm [9].

Our RNA sequencing results in HCL-v cells induced with BAFF suggest that BAFF induces a myriad of genes that are implicated in the chemoresistance of B-cell cancers. In the pathway analysis, NF-κB signaling was one of the most enriched pathways, and many genes that are involved in the nonclassical NF-κB pathway were upregulated. These results also suggest that the effects of BAFF on chemoresistance are potentially mediated through the activation of an alternative NF-κB pathway. Signaling intermediates that were increased include *TRAF1*, *BIRC3*, *NFKB2*, and *RELB*. Additionally, BAFF stimulation also resulted in the upregulation of known anti-apoptotic genes *PIM2* and *BCL-2* that are upregulated with RelB-p52 translocation to the nucleus after pathway activation. For brevity, not all the BAFF-induced genes that are involved in chemoresistance were discussed, but there are many others that were upregulated, including *IL23A, FPR2, INHBA, CCR4, CXCR5*, *JAK3*, *CCR7*, *CCL22*, *TRAF1*, *MAPK8*, *FZD8*, and *RasGRP3* [43,46,52,53,54,55,56,57,58,59,60]. Overall, these findings are supported by recent work in FMS-like tyrosine kinase 3 (FLT3)-mutant acute myeloid leukemia (AML), in which Cao et al. found that nonclassical NF-κB signaling induced the expression of *CXCL1, CXCL5*, and *CXCL8* to promote resistance to gilteritinib [61].

We confirmed the BAFF-mediated protection of patient-derived HCL-v cells from chemotherapy-induced cell death. We have previously demonstrated BAFF-induced chemoprotection in multiple other B-cell cancer models, including acute lymphoblastic leukemia and mantle cell lymphoma, and other groups have also shown this effect in myeloma and chronic lymphocytic leukemia models [26,27,28,29,62]. Given these studies, it was unsurprising that BAFF protects HCL-v, a rare B-cell cancer, from chemotherapy-induced cell death. A more detailed study of BAFF-secreted levels in the HCL-c and HCL-v disease microenvironment will reveal the role of microenvironmental BAFF in chemo-protection, and this will help to design new combination therapies to inhibit BAFF receptors or neutralize BAFF or both in combination with chemotherapy to sensitize drug-resistant cancer cells.

In conclusion, we present evidence that BAFF upregulates multiple genes involved in chemoresistance and functionally decreases sensitivity to cladribine in HCL-v cells. This is likely mediated by BAFF ligation to BAFFR and activation of the nonclassical NF-κB pathway. Because chemosensitivity is a challenge in classical and variant HCL treatment, this study has valuable translational potential. Future work, preclinically or in clinical trials, can investigate combination therapy of PNAs with BAFF neutralization, such as with the BAFF-neutralizing antibody belimumab. Overall, we present an important finding that may lead to new therapeutic combinations that could improve outcomes in HCL patients.

## Figures and Tables

**Figure 1 biomedicines-13-00890-f001:**
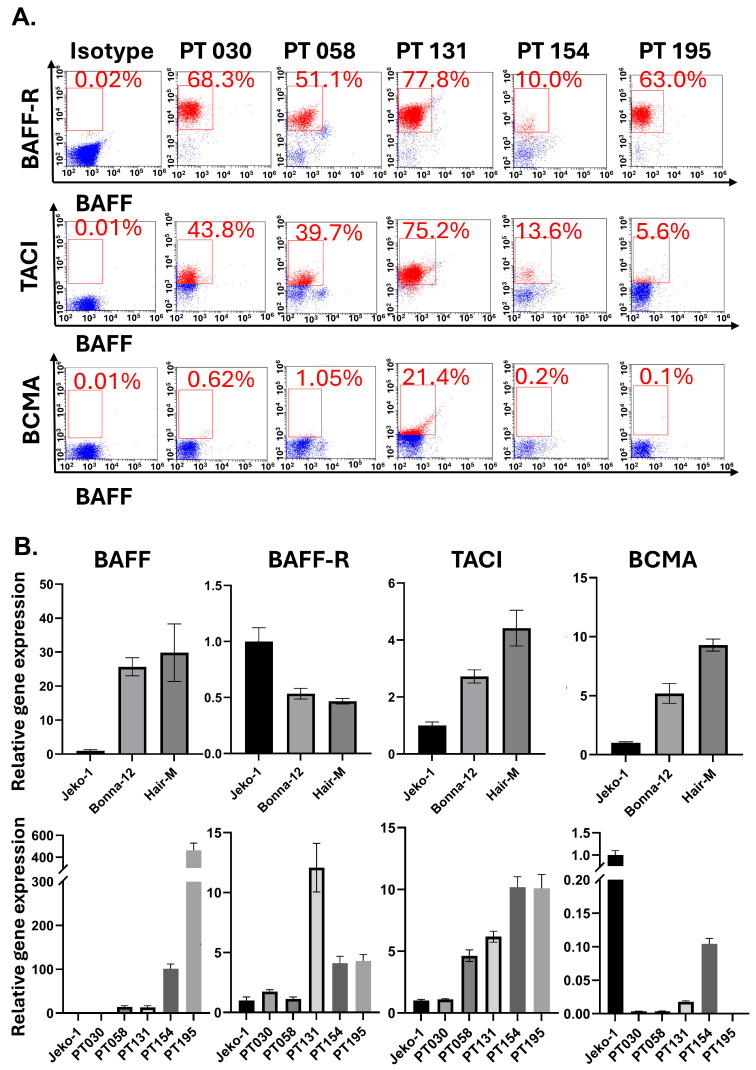
HCL-c/v cells express receptors of BAFF. (**A**) HCL-v patient-derived cancer cells from multiple patients were stained with a panel of flow antibodies: BAFF (APC), BAFF-R (PE), TACI (PE-Cy7), and BCMA (PerCP) or their matched isotype controls. Receptor staining is derived from live/dead gating by forward/side scatter dot plots. Positive populations are relative to the shift of isotype controls. (**B**) The expression of BAFF and its receptors was also analyzed by qPCR with Jeko-1 (mantle cell lymphoma) as the positive control, and 1 μg of cDNA derived from Jeko-1, HCL-v patient-derived cancer cells, or the HCL cell lines Bonna-12 and Hair-M was combined with forward/reverse primers, SYBR green master mix, and nuclease-free water and plated in triplicate. qPCR experiments were repeated for a total of 3 independent experimental replicates.

**Figure 2 biomedicines-13-00890-f002:**
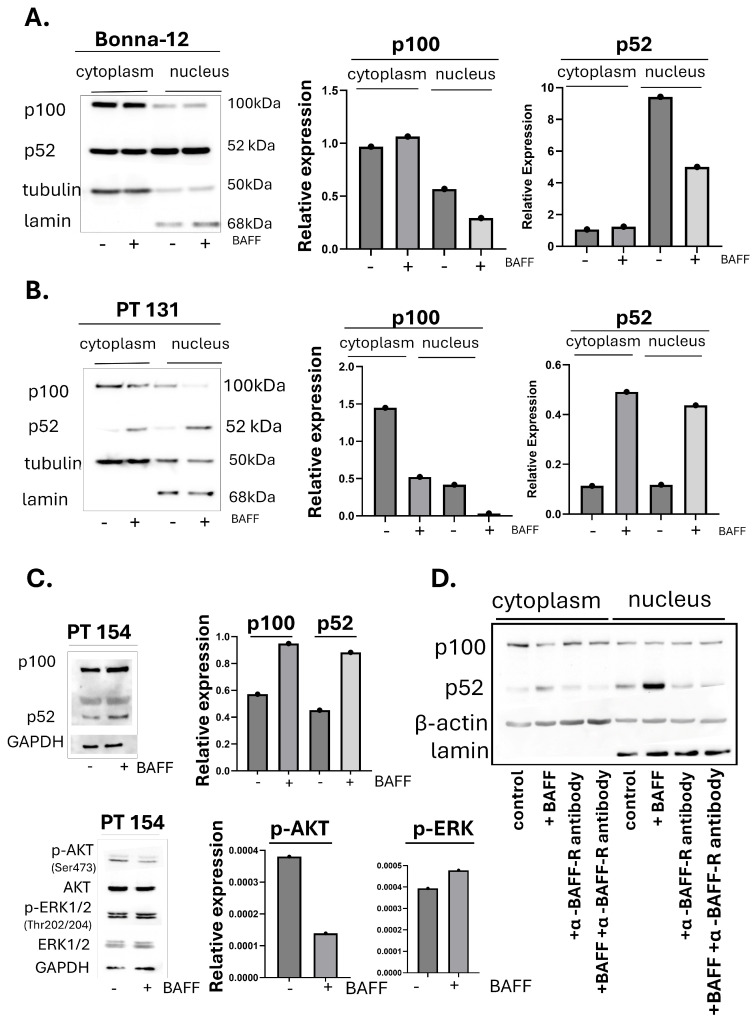
BAFF activates the alternative NF-ĸB pathway in patient-derived HCL-v cells. (**A**) Bonna-12 cells or (**B**,**C**) HCL patient-derived cancer cells (Patients (‘PT’) 131 or 154) were stimulated with 250 ng/mL of recombinant human BAFF. Cytoplasmic and nuclear lysates (Bonna-12 and PT 131) or total cell lysate (PT 154) was harvested after 16 h. Activation of the nonclassical NF-ĸB pathway is measured by the accumulation of p52 (**A**–**C**). AKT and ERK pathway activation is measured in total cell lysate from PT 154 by the accumulation of p-AKT and p-ERK relative to the total protein. Band intensities were measured using ImageJ software. The Bonna-12 Western was performed three independent times, and Westerns using either patient samples were performed at least twice. (**D**) HCL-v patient cells (PT 131) were stimulated with 500 ng/mL with or without 15 µg/mL of BAFF-neutralizing antibody. After 16 h of incubation, cytoplasmic and nuclear protein lysates were harvested, and activation of the nonclassical NF-ĸB pathway was measured by p52 accumulation.

**Figure 3 biomedicines-13-00890-f003:**
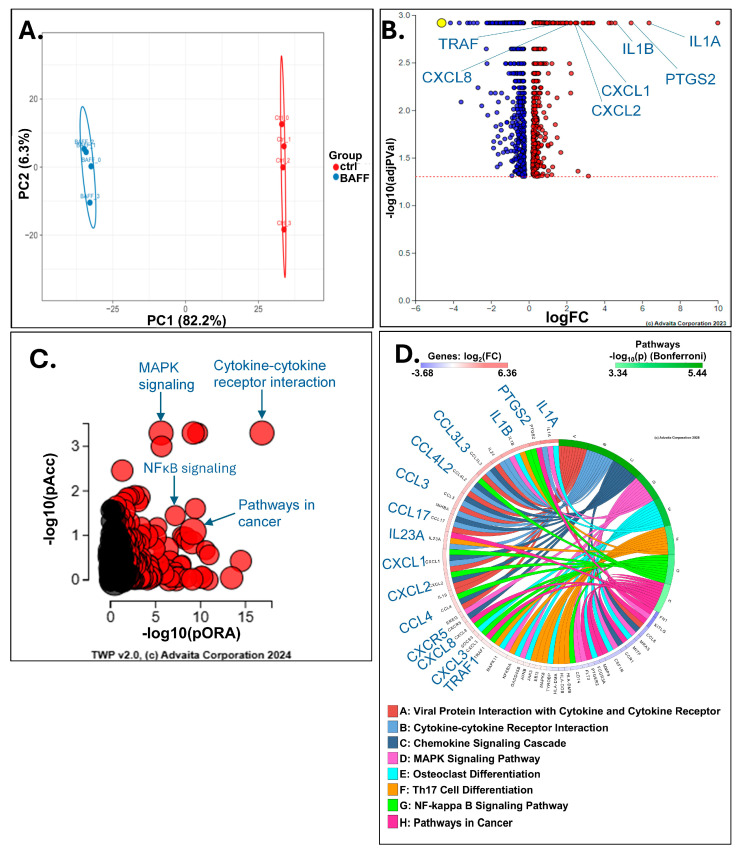
Overview of BAFF induced gene signatures in HCL-v. Cancer cells derived from splenocytes of an HCL patient (Patient 195) were stimulated with 250 ng/mL of recombinant human BAFF for 16 h (*n* = 4 control, *n* = 4 BAFF stimulated). RNA was extracted after incubation and normalized to 50 ng/µL. Sequencing reads were aligned to the human reference genome (GRCh38) using the STAR aligner. Differential genes were identified with a significance cutoff of q-value < 0.05 using Benjamini–Hochberg multiple testing correction. (**A**) Principal component analysis (PCA) shows clustering across quadruplicate samples in each group. (**B**) The volcano plot represents 1447 differentially expressed genes. (**C**) Cytokine–cytokine receptor interaction, MAPK signaling, NF-ĸB signaling, and pathways in cancer were among the most significantly upregulated processes in the pathway analysis. (**D**) The ribbon plot represents the most significantly upregulated genes per process.

**Figure 4 biomedicines-13-00890-f004:**
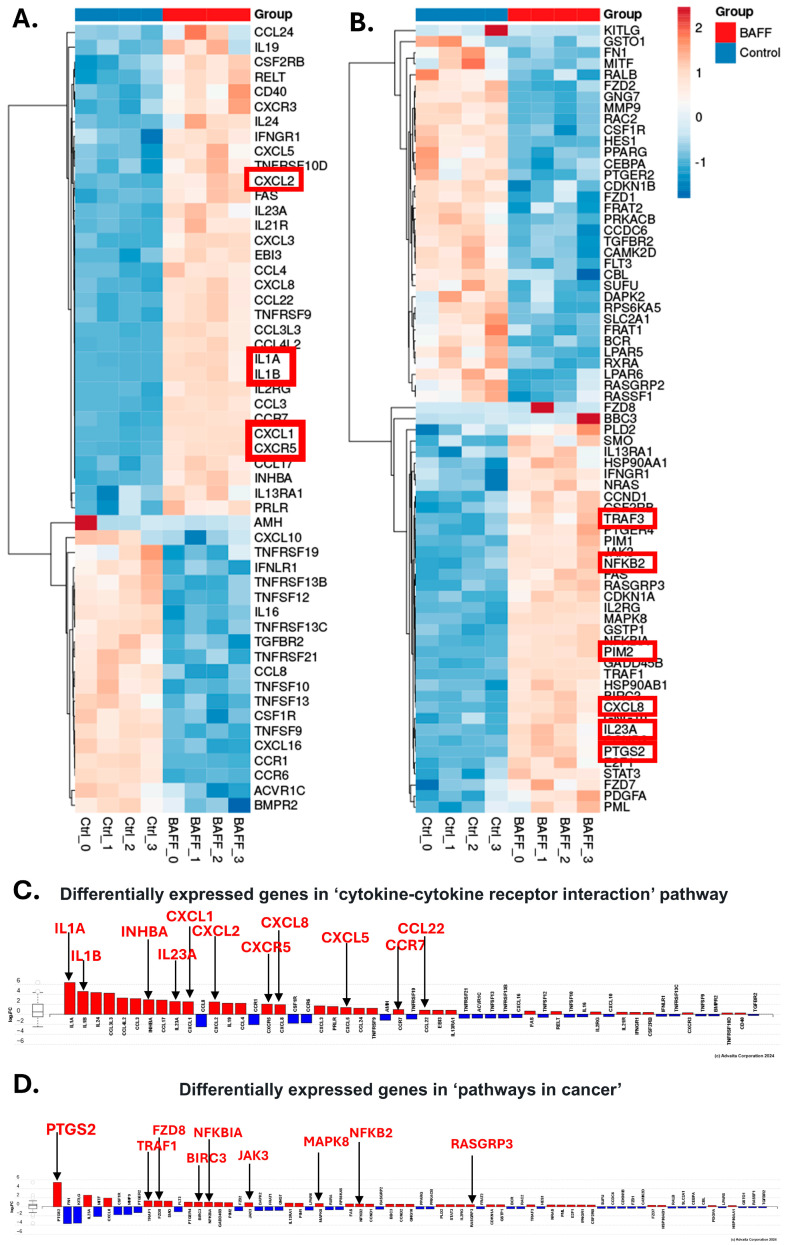
Differentially expressed genes in the processes of ‘cytokine–cytokine’ interaction and ‘pathways in cancer’. Heat maps represent the differentially expressed genes in the pathways (**A**) ‘cytokine–cytokine receptor interaction’ (**B**) and ‘pathways in cancer’ per replicate of each condition. Bar graphs represent fold changes of differentially expressed genes in the (**C**) ‘cytokine–cytokine receptor interaction’ pathways (**D**) and ‘pathways in cancer’.

**Figure 5 biomedicines-13-00890-f005:**
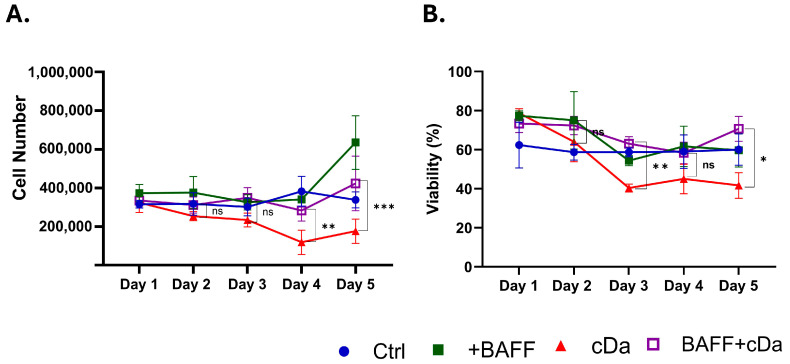
BAFF protects patient-derived HCL-v cells from cladribine. Patient-derived HCL-v cells (Patient 058) were plated at 400,000 cells per well and pre-incubated with 250 ng/mL recombinant human BAFF three days prior to cladribine (cDa) treatment. On day 1, cells were treated with 250 nM cDa and re-stimulated with rhBAFF. Cells were stimulated with BAFF again on day 2. (**A**) Proliferation and (**B**) viability were measured by trypan blue staining. The control well received the solvent control (0.1% DMSO). A one-way ANOVA with Tukey’s multiple hypothesis correction was used to measure significance over time (upper). *p*-values of proliferation (**A**) were 0.0086 (day 4) and 0.0006 (day 5). *p*-values for viability (**B**) were 0.0096 (day 3) and 0.0260 (day 5). The experiment was performed with this patient sample n = 2 times. * *p* < 0.05, ** *p* < 0.01, *** *p* < 0.001.

**Table 1 biomedicines-13-00890-t001:** Primer sequences used for quantitative PCR.

Gene	Primer Sequence
BAFF—forward	GGAGGCAACTCCAGTCAG
BAFF—reverse	CAGTGCAGTCCCAAACTACCAGGACTT
TACI—forward	GAGCAAGGCAAGTTCTATGACC
TACI—reverse	CCTTCCCGAGTTGTCTGAATTG
BCMA—forward	ACCTTGTCAACTTCGATGTTCTT
BCMA—reverse	CAGAGAATCGCATTCGTTCCTT
Actin—forward	TCCACGAAminACTACCTTCAACTC
Actin—reverse	GTCATACTCCTGCTTGCTGAT

## Data Availability

The authors confirm that the data supporting the findings of this study are available within the article.

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
