# Peer review of "B Cell Activating Factor Induces Drug Resistance in Hairy Cell Leukemia Variant"

_biomedicines, 2025, doi:10.3390/biomedicines13040890_

Round 1
Reviewer 1 Report
Comments and Suggestions for Authors
Chemosensitivity is a challenge in the treatment of hairy cell leukemia variant (HCL-v). The authors demonstrate that HCL-v cells derived from multiple patients express B-cell activating factor (BAFF) and its receptors. Recombinant human BAFF stimulates the activation of alternative NF-ĸB and upregulates many genes Involved in chemoresistance in HCL-v cells. Furthermore, BAFF stimulation resulted in functionally decreases sensitivity to cladribine in HCL-v cells.
Highlights: The study is interesting and provides a significant basis for developing novel therapeutic strategies to enhance chemosensitivity to cladribine in HCL-v cells.
However, there are several issues that need to be addressed:
- The authors mention in the manuscript that HCL-v is a rarer form of HCL. However, because it is biologically different from classic HCL, HCL-v is categorized as “unclassifiable splenic B-cell lymphomas” in WHO-HAEM4 and ICC (doi: 10.1038/s41375-022-01764-1) , while it is classified as splenic B-cell leukemia with prominent nucleoli (SBLPN) in the WHO-HAEM5 (doi: 10.1038/s41375-022-01620-2) . To accurately define HCL-v, it is recommended that the authors consult updated and high-quality references. Furthermore, to avoid using HCL and HCL-v interchangeably and causing confusion, it is recommended that the authors review the entire manuscript to ensure that the expression is used correctly and clearly.
- It is recommended to show the expressions results of BAFF and its receptors in all HCL-v patients.
- In the Flow Cytometry subsection, what tissues are the patients’ cells derived from?
- In the Figure 2C, a letter “F”is missing from the right image.
- It is recommended to spell out the full term for any abbreviations when they first appear in the text, including but not limited to cDa and CAML.
Author Response
- The authors mention in the manuscript that HCL-v is a rarer form of HCL. However, because it is biologically different from classic HCL, HCL-v is categorized as “unclassifiable splenic B-cell lymphomas” in WHO-HAEM4 and ICC (doi: 10.1038/s41375-022-01764-1), while it is classified as splenic B-cell leukemia with prominent nucleoli (SBLPN) in the WHO-HAEM5 (doi: 10.1038/s41375-022-01620-2). To accurately define HCL-v, it is recommended that the authors consult updated and high-quality references. Furthermore, to avoid using HCL and HCL-v interchangeably and causing confusion, it is recommended that the authors review the entire manuscript to ensure that the expression is used correctly and clearly. We agree with the reviewer on this important point and carefully revised the article to differentiate between the two forms of hairy cell leukemia throughout the manuscript. The changes that we made are as follows. In the abstract, we removed the language describing that HCL-v is a rarer form and instead describe HCL-v as a biologically distinct entity from HCL-c. To further elaborate, in the introduction when HCL-v is first described, we now include the classification from the 5th edition WHO-HAEM as a splenic B-cell leukemia with prominent nucleoli. This is now included as reference #15. Further, When HCL is described in the introduction, we specify when we are discussing specifically classical HCL (see paragraph 1). Further, generally, anywhere in the manuscript where HCL is mentioned, it is specifically described as HCL-c or HCL-v. We more carefully have described our results in the context of HCL-v when HCL-v patient samples were used, except where the HCL-c cell lines (Bonna-12, Hair-M) were used (Figure 1B, 2A).
2. It is recommended to show the expressions results of BAFF and its receptors in all HCL-v patients.
To address this suggestion, we now include both the flow staining and qPCR of BAFF and receptor expression for all the patient-derived samples used in this study. We have updated Figure 1 with these new figures.
3. In the Flow Cytometry subsection, what tissues are the patients’ cells derived from?
To address this concern, we have included a new section under methodology that describes the origin of the patients’ cells used throughout this study. See “2.1 Patient Samples.” These de-identified patient samples are derived from splenic lymphocytes of HCL-v patients. These cells were provided from the Wilmont Cancer Institute at the University of Rochester. They are discarded samples deemed unnecessary for clinical use and obtained by an approved IRB-exemption protocol.
Comments on the Quality of English Language
1. In the Figure 2C, a letter “F”is missing from the right image.
We thank the reviewer for pointing this out. We edited the figure, so that ‘BAFF’ is not cut off.
2. It is recommended to spell out the full term for any abbreviations when they first appear in the text, including but not limited to cDa and CAML.
Upon this suggestion, we have carefully reviewed the text and ensured that any abbreviations used were defined when first introduced. We have modified the text so that we included the abbreviation “cDa” when cladribine first appears in the text for consistency (lines 21, 61) and wrote out the full name of TACI as transmembrane activator and calcium-modulator and cyclophilin ligand (CAML) interactor (line 93-94). Other errors that we corrected include the abbreviations for classical hairy cell leukemia (HCL-c) and variant hairy cell leukemia (HCL-v) in the abstract, lines 20 and 23, respectively. Additionally, the full name of NF-ĸB was written out when it first appeared in the abstract and introduction (lines 29 & 95-96) and tumor necrosis factor (TNF) was written out in the introduction (line 90). In the introduction we again include the abbreviation cDa the first time that cladribine is introduced (line 61). The full name of MAP kinase, mitogen-activated protein kinase, was elaborated in line 56-57.
Reviewer 2 Report
Comments and Suggestions for Authors
Fritz et al identified the cladribine-resistance mechanism in vitro leukemia models. Through multiomic evidence, they revealed the involvement of BAFF and BAFF-R mediated non-canonic NF-kB pathway in chemoresistance of HCL and HCL-v cells. It’s well-written manuscript. I have the following comments:
Please provide clinical profiles of patients whose samples were used in this study, and IRB numbers. Also, please provide the basic information of the cell line used in this study.
For the flow analysis of patient samples, how did you gate the viable cells from dead/dying cells?
Have you examined the effect of anti-BAFF-R antibody? Did you see any changes of non-classical NF-kB pathway and cyto/chemokines/receptors?
In a previous study of acute myeloid leukemia, the increased expression of NF-kB2, CXCL1/CXCL5/CXCL8, CXCR2 and their mediated TKI-resistance has been reported. To support your findings in hair cell/-v leukemia, please cite this study PMID: 35625776 or DOI: 10.3390/biomedicines10051038.
Author Response
Please provide clinical profiles of patients whose samples were used in this study, and IRB numbers. Also, please provide the basic information of the cell line used in this study.
To address this concern, we have included two new sections in the methodology: “2.1 Patient Samples” and “2.2 Cell Lines.” This reads as follows: “2.1 Patient Samples. All HCL-v patient samples were provided by the Hematological Malignancy Procurement Program of the Wilmot Cancer Institute at the University of Rochester Medical Center. These are splenic lymphocytes obtained by the surgical pathology department after diagnostic biopsies. These discarded and de-identified patient samples were obtained under an approved IRB-exemption protocol. All patients had an HCL-v diagnosis and expressed the markers CD11c, CD19, CD20, and CD22.” “2.2 Cell lines. The Bonna-12 (Item #: ACC 150) and Hair-M (Item #: ACC 762) cell lines were obtained from the Leibniz Institute DSMZ - German Collection of Microorganisms and Cell Cultures. Both cell lines were originally established from hairy cell leukemia patients.”
For the flow analysis of patient samples, how did you gate the viable cells from dead/dying cells?
The viable cells were gated on by forward/side scatter flow dot plots for each sample. Only live cells were included in the receptor expression analysis. An example of the gating strategy is included below (please see the word document attached). We have now included this gating strategy within the methodology, so readers know how we gated for viable cells (see lines 140-142). This is also updated in the figure legend (see lines 240-41).
Have you examined the effect of anti-BAFF-R antibody? Did you see any changes of non-classical NF-kB pathway and cyto/chemokines/receptors?
In response to this question, we treated HCL-v cells (Patient 131) with a BAFF-receptor-neutralizing antibody before BAFF stimulation and harvested cytoplasmic and nuclear protein fractions after 16 hours. We then looked for activation of the nonclassical NF-ĸB pathway and compared activation with the BAFF-activated control. We found that BAFF Receptor blockade resulted in significantly reduced activation of the non-classical NF-ĸB pathway, supporting our conclusion that BAFF-mediated chemoprotection occurs through BAFF-Receptor activation of this pathway. We have added these findings as panel D of Figure 2 and added a description of these results to the methods, results, and discussion sections (lines 182-187, 269-273, 390-392, respectively). We took the remaining cells that were not used for protein lysates for flow staining of chemokine receptors and did not see significant changes in protein expression at this time point, though CXCR2, CXCR3, and CXCR5 expression trended toward decreasing with addition of the BAFF-R neutralizing antibody (data in the word document attached).
In a previous study of acute myeloid leukemia, the increased expression of NF-kB2, CXCL1/CXCL5/CXCL8, CXCR2 and their mediated TKI-resistance has been reported. To support your findings in hair cell/-v leukemia, please cite this study PMID: 35625776 or DOI: 10.3390/biomedicines10051038.
We thank the reviewer for bringing this article to our attention as we did not see it upon the first draft of the manuscript. A reference to this article is now included after the RNA sequencing component of our discussion section (reference number 62). This now reads: “Overall, these findings are supported by recent work in FMS-like tyrosine kinase 3 (FLT3)-mutant acute myeloid leukemia (AML), in which Cao et al found that nonclassical NF-ĸB signaling induced expression of CXCL1, CXCL5, and CXCL8 to promote resistance to gilteritinib.” (lines 420-424).

Round 2
Reviewer 1 Report
Comments and Suggestions for Authors
The authors have addressed my all queries. I have no further comments.